# Kinematic analysis of the underwater glide phase during the push-off in competitive swimmers

Alfonso Trinidad[1,2]*

1 Departamento de Educación e Innovación Educativa, Facultad de Ciencias Sociales, Universidad Europea de Madrid, Villaviciosa de Odón, Madrid, Spain, 2 Aqualab Research Group, Universidad Europea de Madrid, Villaviciosa de Odón, Madrid, Spain

* alfonso.trinidad@universidadeuropea.es

**Data Availability Statement:** All relevant data are within the manuscript and its Supporting information files.

**Funding:** The author(s) received no specific funding for this work.

## Abstract

The aim of the study was to analyze the kinematic parameters of the push-off start during the underwater glide in competitive swimmers. 74 swimmers participating in the Spanish Championships were filmed and analyzed by DLT-2D photogrammetry after the push-off start in crawl, backstroke and butterfly. Between genders there were differences in distance and speed. Male swimmers covered greater distances (1.37±0.06 vs 1.20±0.05 m, $\eta 2$ = 0.02, F = 3.85, p = 0.05) and were faster (2.36±0.03 and 2.08±0.03 m/s, $\eta 2$ = 0.14, F = 36.14, p<0.001) than female swimmers. Between strokes there were greater differences in time ($\eta 2$ = 0.06, F = 6.76, p = 0.00) and distance ($\eta 2$ = 0.38, F = 67.08, p< 0.001), than in speed ($\eta 2$ = 0.05, F = 5.63, p< 0.001). During the backstroke, less time (0.50±0.04 s) and distance (1.01±0.07 m) were used, being the slowest style (2.12±0.04 m/s). In butterfly, less time (0.63±0.04 s) and distance (2.25±0.04 m) were used, while crawl was the fastest (2.30 ±0.04 m/s). These results allow us to characterize the underwater glide phase and provide useful data for both competitive swimmers and coaches to improve performance.

## Introduction

Underwater swimming or the fifth stroke occupies a remarkable place in competitive swimming and is characterized by high speeds reached compared to surface swimming due to lower resistance [1] and the fast initial velocities achieved when diving (start) or pushing off the wall (turn) [2]. Although it could be described as a summative phase made up of different phases (gliding, undulating underwater movement and transition to swimming), or even divided into sub-phases [3], it is preceded by another non-underwater phase (surface swimming phase) which depends on the progression developed in the underwater sections.

The first of these phases is the glide, moment in the swimmer tries to maintain speed without any action to propel the body. For its correct execution it is necessary to achieve and maintain a hydrodynamic, horizontal and progressive position in the water for as long as possible [4–7], in order to maintain the speed acquired after the impulse of the start or turn [8]. Therefore, it occupies between 10 and 25% of the total time of the test [9] or between 18 and 28% of

**Competing interests:** The authors have declared that no competing interests exist.

the start time [10], becoming a transitional phase before the undulating underwater movement [4], which is crucial to achieve a good performance in the whole underwater swimming.

Researches that have analyzed different kinematic parameters of the glide after the start and the turn have tried to give an answer to the so-called glide efficiency, or the ability to maintain speed and minimize deceleration over time [11]. However, most of these studies, after analyzing the start, provide different data according to the underwater reference taken (5, 7 and 10 m) [11, 12] or the swimming style before the glide [13], without clarifying at what speed the swimmer should glide [14], for how long [15] or at what distance the swimmer should stop this phase [16] and start the underwater leg kick. We even compared different types of starts [12], different from those currently used on the Omega OSB11. Finally, we have compared the high-speed swimsuit and the normal swim [17], without taking into account the current regulations for the use of these swimsuits [18].

On the other hand, studies that have focused on analyzing the effect of gliding after turns have shown little difference in glide time and distance [19, 20], suggesting that swimmers used different strategies during the turn approach and thrust speed, thus reducing glide distance and resistance generated by a rapid incorporation of underwater wave motion [21]. Data on glide distance were even found to be lower in studies using monofins [22] than in other studies without aids after turning [19, 20]. There are also some discrepancies as to what would be the ideal glide speed, between 0.90 and 2 m/s [15, 23] or between 1.9 and 2.2 m/s [4, 24].

Therefore, based on the literature reviewed, the aim of the present study was to analyse the kinematic parameters of the push-off start during the underwater glide phase in competitive swimmers at national level. It was hypothesised that the kinematic parameters during the push-off start would depend on swimming style and gender.

## Materials and methods

### Participants

A total of seventy-four swimmers, 33 males (16.5±1.2 years old and 62.3±7.2 kg) and 41 females (15.5±1.2 years old and 50.5±6.5 kg) from eight clubs of the Madrid Swimming Federation (Specialised Centre of Sport Technification in Swimming "M-86" of Madrid; Alcobendas Swimming Club; Gredos San Diego Swimming Club of Moratalaz; Jimenez Swimming Club; Pozuelo Swimming Club; Real Canoe Swimming Club; Club A.D. Rivas Swimming and SAFA Swimming Madrid) voluntarily participated in the study. All swimmers belonged to the national junior category and participated in the Spanish Championships. According to the recent standardisation of swimming performance proposed by Ruiz-Navarro et al. [25], the study subjects would be identified as level two and three swimmers. Their training programmes consisted of between 12 and 20 hours of water training and between 3 and 6 hours of dry training per week. Inclusion criteria for the experiment included continuous training attendance (9 water sessions per week) and the absence of any major injury in the three months prior to the experiment. Swimming experience was 7.1±2.3 years for males and 5.9 ±1.6 years for females. Participants (>18 years) or their legal guardians (<18 years) signed a written consent document to participate in the study between 27 and 29 June 2022. The document was approved by the local ethics committee (number 2020–082) in accordance with the Declaration of Helsinki [26]. The study was conducted at the Specialised Centre of Sport Technification in Swimming "M-86" in the Community of Madrid between 4 and 6 July 2022. All participants were assessed individually and randomly. The age, weight and swimming experience of each swimmer were recorded. They then underwent a standardised warm-up of approximately 1.200 m and were given instructions about the test. Finally, they were filmed

while performing the test in the order of their preferred stroke (crawl or butterfly or backstroke) and in the middle lane of the 50 x 25 m indoor pool.

## Instruments

Two JVC GY-DV500E video cameras (located in the underwater windows at the side of the pool) recorded the swimmers at 25 Hz. The distance between the lenses of these cameras and the swimmers was approximately 12.5 m (Fig 1). Using DLT-2D algorithms [27], the actual coordinates (x, y) were calculated from the pixel coordinates digitised on the computer screen (Photo23D, Universidad Politécnica de Madrid, Spain) [28]. A calibration frame consisting of four points with known coordinates on a vertical plane in the swim lane was used to calculate the coordinates. The mechanical model used to determine the swimmer's position was that of a point represented by the swimmer's hip [29], from which the coordinates (x, y) and the study variables were obtained.

## Design and procedures

Swimmers were filmed individually underwater in three events (butterfly, backstroke and crawl, in random order) from the wall push to the 15 m mark at maximum swimming speed, with a period of free swimming underwater in which only the underwater glide phase to the surface was analysed. The breaststroke was not filmed during the race because the swimmers had to finish the glide phase with the dolphin kick. This phase started from the moment when the swimmer's feet separated from the wall until the moment when the ankle joint started the downstroke (butterfly and crawl) or the upstroke (backstroke). Once the coordinates (x, y) of the athlete's position in each time interval were obtained, the following variables could be

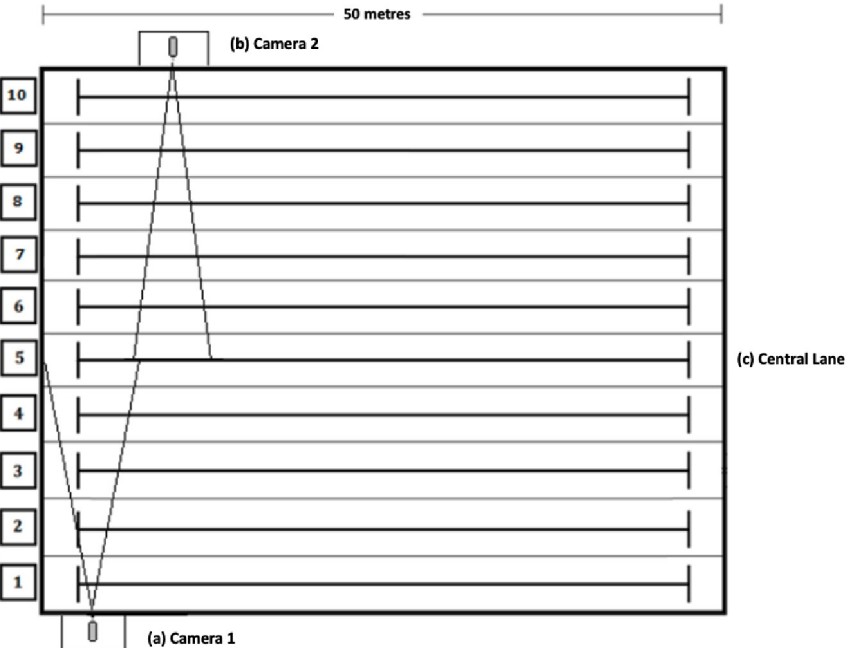

**Fig 1. Diagram showing camera positioning and location.** (a) Camera 1 in underwater window n˚ 1; (b) Camera 2 in underwater window n˚ 2; (c) Central lane of the 50 x 25 m indoor pool.

obtained: time (T_DES), distance (D_DES) and velocity (V_DES) during the gliding phase (Table 1).

## Data analysis

Statistical analysis of the data was performed using SPSS version 29.0 (SPSS Inc., Chicago, IL, USA), with means and standard deviations calculated for each of the kinematic variables (time, distance and speed). A repeated measures analysis of variance was then performed with an between-subjects factor (gender) and an within-subjects factor (style). The F-value contrast was used together with the degrees of freedom for the statistical probability of the effect of a factor on the population variance. The significance level (p-value) was set at 0.05. Finally, the results were interpreted in terms of effect size using eta-squared ($\eta2$) values according to thresholds representing small, moderate, large, very large and near perfect correlations, which were 0.1, 0.3, 0.5, 0.7 and 0.9 respectively [30].

## Results

Comparison between the genders showed that males covered more distance during the slide than females ($\eta2 = 0.02$, F = 3.85, p = 0.05), even at a higher velocity ($\eta2 = 0.14$, F = 36.14, $p < 0.001$). However, there were no significant differences in gliding time between the genders ($\eta2 = 0.00$, F = 0.03, p = 0.87). Descriptive scores and multivariate analysis for each group are shown in Table 2.

When comparing the strokes, significant differences were found in time ($\eta2 = 0.06$, F = 6.76, p = 0.00), distance ($\eta2 = 0.38$, F = 67.08, $p < 0.001$) and glide velocity ($\eta2 = 0.05$, F = 5.63, $p < 0.001$). During the push-off start in the backstroke, the swimmers spent less time developing the glide (0.50±0.04 s) and covered less distance (1.01±0.07 m), in addition to being the slowest compared to the other strokes (2.12±0.04 m/s). On the other hand, butterfly swimmers took less time (0.63±0.04 s) and covered less distance (2.25±0.04 m). However, the crawl stroke was the fastest of the three strokes (2.30±0.04 m/s) (Table 3).

## Discussion

The aim of the present study was to analyse the kinematic parameters of the push-off start during the underwater glide phase in competitive swimmers at national level.

The gender comparison during the underwater glide phase showed that the time taken by the swimmers was the same in both cases, although the males glided and swam faster after the push-off start (≈12%) due to a greater impulse from the wall and better body shape and hydrodynamics, which may have provided them with less resistance [31, 32] to achieve an efficient underwater glide [7]. These data differ from those of Puel et al. [19, 20] in terms of glide time,

**Table 1. Variables measured during the underwater gliding phase.**

| Phase | Code | Description |
|---|---|---|
| **Underwater glide time (s)** | T_DES | Time in seconds from the moment when the swimmer's feet leave the wall in the ventral position (crawl and butterfly) or dorsal position (backstroke) to the moment when the ankle joint begins the downstroke (crawl and butterfly) or upstroke (backstroke). |
| **Underwater glide distance (m)** | D_DES | The horizontal distance in metres from the hip position at the moment when the swimmer's feet leave the wall in the ventral (crawl and butterfly) or dorsal (backstroke) position, to the moment when the ankle joint begins the downstroke (crawl and butterfly) or upstroke (backstroke). |
| **Underwater glide velocity (m/s)** | V_DES | The average velocity of the swimmer's movement in m/s during the underwater glide phase (crawl, backstroke and butterfly). |

**Table 2. Descriptive results and multivariate analysis of kinematic parameters, glide time (T_DES), glide distance (D_DES) and glide velocity (V_DES) in national level competitive swimmers as a function of gender.**

|  | Males | Females | Sum of squares | gl | Mean square | F | Sig. | η2 |
|---|---|---|---|---|---|---|---|---|
| **T_DES (s)** | 0.59±0.03 | 0.59±0.03 | 0.00 | 1 | 0.00 | 0.03 | 0.87 | 0.00 |
| **D_DES (m)** | 1.37±0.06 | 1.20±0.05 | 1.37 | 1 | 1.37 | 3.85 | 0.05* | 0.02 |
| **V_DES (m/s)** | 2.36±0.03 | 2.08±0.03 | 4.27 | 1 | 4.27 | 36.14 | <0.001* | 0.14 |

* Significant differences at $p < 0.05$.

although their swimmers aimed to reduce their distance so as not to increase glide time, drag and energy expenditure and thereby cause a loss of speed [4]. However, these authors analysed the turning phase, and glide time may have benefited from the approach phase, unlike the present study, which only included the wall thrust. In terms of glide distance and speed, the data from this study are consistent with those from other studies [19, 20, 33], although their recordings were higher. However, the main difference was observed in the performance level of the swimmers analysed (sub-elite and elite vs. national). Therefore, it could be speculated that high level swimmers with good hydrodynamic position and body alignment [34] may be able to minimise their time and increase their glide distance. It could also be argued that in order to perform a correct glide, a strategy could be adopted during the turn [19, 20]. Or even, in order to reduce their drag and not lose speed, to make a quick incorporation into the underwater kick [21]. Therefore, these factors may have been conditioning factors in the young swimmers in the study, due to a possible lack of specific work in the water. However, other studies that have limited the glide distance to 3, 5 and 10 m [11, 33], together with different analysis conditions (start and turn vs. push-off start), have recorded lower speed values, which do not show real data that can be extrapolated to competitions, as swimmers rely more on their senses to finish the glide and start the underwater kick. However, further research in this area would be needed to analyse other swimming events and to provide data to compare men and women with the same level of performance.

The comparison between styles showed that the times recorded were lower than those of Sanders [35], due to the advantage of entering the water after the impulse from the starting podium and not after the push-off start. In contrast, other results in the literature have been lower than ours and have only focused on analysing a position and style after the start [3, 22] and the turn [19–22], with few studies analysing swimming styles after the push-off start. Only the times of Hay [15] and Wada et al. [17] were close to the crawl and butterfly times analysed, together with those of Pearson et al. [12], which are in the same swimming category despite the methodological differences (start vs. push-off start). In terms of distance, the results of other studies analysing gliding and turning with the pronated monofin [22] showed lower values than ours, because the swimmers performed a short propulsion, thus reducing their gliding

**Table 3. Descriptive results and multivariate analysis of kinematic parameters, glide time (T_DES), glide distance (D_DES) and glide velocity (V_DES) in national level competitive swimmers as a function of swimming style.**

|  | Crawl | Backstroke | Butterfly | Sum of squares | gl | Mean square | F | Sig. | η2 |
|---|---|---|---|---|---|---|---|---|---|
| **T_DES (s)** | 0.66±0.04 | 0.50±0.04 | 0.63±0.04 | 1.15 | 2 | 0.57 | 6.76 | 0.00* | 0.06 |
| **D_DES (m)** | 1.47±0.07 | 1.01±0.07 | 1.37±0.07 | 8.77 | 2 | 4.39 | 13.59 | <0.001* | 0.11 |
| **V_DES (m/s)** | 2.30±0.04 | 2.12±0.04 | 2.25±0.04 | 1.48 | 2 | 0.74 | 5.63 | 0.00* | 0.05 |

* Significant differences at $p < 0.05$.

distance. On the other hand, other studies on elite swimmers have found greater distances between strokes [19, 33] compared with our national-level swimmers, despite the fact that the references analysed were different (distance of the turn between the wall and the head vs. distance after the push-off start to the hip), with the level of performance being the main cause. Finally, with regard to glide speed, the data from other studies varied according to the type of study (start, turn, towing, etc.) [3, 4, 8, 17, 24]. However, only the speeds found in a study analysing crawl turns [19] were close to ours, and other studies have shown even higher values, regardless of the category of swimming analysed (national, sub-elite and elite) [14, 33]. Therefore, it could be confirmed that the orientation of the style affects the speed after the push-off start.

In this line, and with regard to the differentiation between styles, the backstroke was the one that covered the least time (6.4%) and the least distance during the glide (9%) and, consequently, the one that registered the lowest speed due to the development of the phase in the dorsal position, according to the results obtained in drag tests at different speeds and positions to quantify passive resistance [36]. All this could have led to a higher resistance rate compared to movements in the ventral position, since for Lyttle et al. [4] the minimum drag was found in the ventral hydrodynamic position, regardless of speed, coinciding with the highest times of the crawl and butterfly strokes analysed, together with their postural development in the ventral position. Therefore, it could be speculated that postural control in the dorsal position was much more difficult or not sufficiently hydrodynamic in terms of linear adjustment of the trunk and upper body segments [27, 37, 38]. Consequently, it may have caused an increase in resistance and a loss of speed after the push-off start, making it the slowest style, although our data were close to those of De Jesus et al. [13], mainly due to the similarity of the push-off start after the backstroke. Nevertheless, this may have forced the swimmers in the study to shorten the backstroke phase and start the underwater kick.

## Conclusions

The kinematic study of the gliding phase after the push-off start showed that male swimmers covered more distance than female swimmers, due to their greater speed after the push-off start. However, both sexes used similar times in the development of the gliding phase, with minimal differences between them. On the other hand, in the national level swimmers, the underwater glide phase was much faster and a greater distance was covered in the crawl and butterfly strokes, compared to the backstroke. However, the slower speed in the backstroke was due to the shorter time and distance covered during the glide. These results allow us to characterize the gliding phase after the wall push when comparing between swimming styles and between genders, the latter variable being little studied in the literature, as well as the conditions of this phase. In addition, the present study could provide useful information to coaches to improve the performance of competitive swimmers.

## Supporting information

**S1 File.**
(XLSX)

## Acknowledgments

This study was supported by eight clubs of the Madrid Swimming Federation (Specialised Centre of Sport Technification in Swimming "M-86" of Madrid; Alcobendas Swimming Club; Gredos San Diego Swimming Club of Moratalaz; Jimenez Swimming Club; Pozuelo

Swimming Club; Real Canoe Swimming Club; Club A.D. Rivas Swimming and SAFA Swimming Madrid).

## Author Contributions

**Conceptualization:** Alfonso Trinidad.

**Data curation:** Alfonso Trinidad.

**Formal analysis:** Alfonso Trinidad.

**Funding acquisition:** Alfonso Trinidad.

**Investigation:** Alfonso Trinidad.

**Methodology:** Alfonso Trinidad.

**Project administration:** Alfonso Trinidad.

**Resources:** Alfonso Trinidad.

**Software:** Alfonso Trinidad.

**Supervision:** Alfonso Trinidad.

**Validation:** Alfonso Trinidad.

**Visualization:** Alfonso Trinidad.

**Writing – original draft:** Alfonso Trinidad.

**Writing – review & editing:** Alfonso Trinidad.

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
