## [Decision Letter · Decision Letter 0]

2 Nov 2023

PONE-D-23-27558Kinematic analysis of the underwater glide phase during the push-off in competitive swimmersPLOS ONE

Dear Dr. Trinidad Morales,

Thank you for submitting your manuscript to PLOS ONE. After careful consideration, we feel that it has merit but does not fully meet PLOS ONE’s publication criteria as it currently stands. Therefore, we invite you to submit a revised version of the manuscript that addresses the points raised during the review process.

We look forward to receiving your revised manuscript.

Kind regards,

Efrem Kentiba, PhD

Academic Editor

PLOS ONE

Journal Requirements:

"This study was supported by the Aqualab Research Group of Universidad Europea de Madrid (Spain) and the Universidad Politécnica de Madrid (Spain)."

6. Please ensure that you include a title page within your main document. You should list all authors and all affiliations as per our author instructions and clearly indicate the corresponding author.

7. Please include a caption for figure 1.

**Additional Editor Comments:**

When reading the manuscript, I do not identify it as a study of potential interest to professionals and academics in its current status. Besides, merely descriptive. So, can you run additional analysis to show some relationships?

Reviewers' comments:

Reviewer's Responses to Questions

**Comments to the Author**

1. Is the manuscript technically sound, and do the data support the conclusions?

Reviewer #1: Yes

Reviewer #2: Yes

2. Has the statistical analysis been performed appropriately and rigorously? 

Reviewer #1: Yes

Reviewer #2: Yes

3. Have the authors made all data underlying the findings in their manuscript fully available?

Reviewer #1: Yes

Reviewer #2: Yes

4. Is the manuscript presented in an intelligible fashion and written in standard English?

Reviewer #1: Yes

Reviewer #2: Yes

5. Review Comments to the Author

Reviewer #1: The abstract written in a well manner and included all the important points.

The introduction written in a proper way; briefly addresses all the necessary points including the importance of the topic and the research as a whole.

The conclusion developed properly, discussed the findings and the conclusion drawn properly.

The total population, sample size and sampling methods should be written and explained explicitly.

The hour of training is different. This may have a great impact on the performance of an athlete and comparing athletes which train differently may bring different results.

Explanatory statements at the beginning of the abstract are not as such important. So it had better to delete the statement from line 4-9.

Generally I believe the article may contributes something to the knowledge of the competitive athletes and their coach to address each and every activity that have an impact on the performance of an athletes during training and competition in particular and to the sports science in general.

Reviewer #2: I read this work with interest and in my opinion authors did a good article. Some deficiencies are given under these lines, but an extensive review is unnecessary.

INTRODUCTION

Well written, but too little introduction to the topic and too much discussion and scientific examples. Much part of the text is suitable for discussion. Please add some basic information about underwater swimming, glides, breakouts, differences in glides/strokes etc.

MATERIAL AND METHODS

L77-78 : Consistently one or two digits after the decimal point. I advise one.

L86-87 : Add some infromation about season period (kilometers, intesity, hours etc.)

DESIGN AND PROCEDURES

L104-105 : If breaststroke was not in research, give some information in introduction that this stroke and glide is much different from others.

L111-114 : Please use one term – velocity or speed. Add SI units to all.

RESULTS

L136 : Table 2. Add SI Units in first column (also Table 3). The female and male were exactly the same results in time [s] ?

DISCUSION

The discussion is well written, but with fragments that are not suitable for publication because the number of results is too detailed. Less details and more typical conclusions please. I strongly recommend that you should use a typical thesis conclusion and compare it with other researchers. Especially in conclusion section, You should not use exact results, numbers and units etc. – only fruits of Your labour.

6. PLOS authors have the option to publish the peer review history of their article (what does this mean?). If published, this will include your full peer review and any attached files.

Reviewer #1: No

Reviewer #2: No

---

## [Author Response · Author response to Decision Letter 0]

16 Nov 2023

All responses to reviewers' and editor's comments and suggestions have been attached.

---

## [Decision Letter · Decision Letter 1]

6 Dec 2023

Kinematic analysis of the underwater glide phase during the push-off in competitive swimmers

PONE-D-23-27558R1

Dear Dr. Morales

We’re pleased to inform you that your manuscript has been judged scientifically suitable for publication and will be formally accepted for publication once it meets all outstanding technical requirements.

Kind regards,

Efrem Kentiba, PhD

Academic Editor

PLOS ONE
---

## [Editor Report · Acceptance letter]

20 Dec 2023

PONE-D-23-27558R1 

PLOS ONE

Dear Dr. Trinidad Morales, 

I'm pleased to inform you that your manuscript has been deemed suitable for publication in PLOS ONE. Congratulations! Your manuscript is now being handed over to our production team.

Kind regards, 

on behalf of

Dr. Efrem Kentiba 

Academic Editor

PLOS ONE